# Inhibition of Toll-like Receptors Alters Macrophage Cholesterol Efflux and Foam Cell Formation

**DOI:** 10.3390/ijms25126808

**Published:** 2024-06-20

**Authors:** Jaemi Kim, Ji-Yun Kim, Hye-Eun Byeon, Ji-Won Kim, Hyoun-Ah Kim, Chang-Hee Suh, Sangdun Choi, MacRae F. Linton, Ju-Yang Jung

**Affiliations:** 1Department of Rheumatology, School of Medicine, Ajou University, Suwon 16499, Republic of Korea; funny1126@hotmail.com (J.K.); chsuh@ajou.ac.kr (C.-H.S.); 2Institute of Medical Science, School of Medicine, Ajou University, Suwon 16499, Republic of Korea; 110306@aumc.ac.kr (J.-Y.K.);; 3Department of Molecular Science and Technology, Ajou University, Suwon 16499, Republic of Korea; sangdunchoi@ajou.ac.kr; 4Department of Medicine, Cardiovascular Medicine, Vanderbilt University Medical Center, Nashville, TN 37232, USA; macrae.linton@vumc.org; 5Department of Pharmacology, Vanderbilt University, Nashville, TN 37235, USA

**Keywords:** atherosclerosis, toll-like receptor, cholesterol efflux, foam cells

## Abstract

Arterial macrophage cholesterol accumulation and impaired cholesterol efflux lead to foam cell formation and the development of atherosclerosis. Modified lipoproteins interact with toll-like receptors (TLR), causing an increased inflammatory response and altered cholesterol homeostasis. We aimed to determine the effects of TLR antagonists on cholesterol efflux and foam cell formation in human macrophages. Stimulated monocytes were treated with TLR antagonists (MIP2), and the cholesterol efflux transporter expression and foam cell formation were analyzed. The administration of MIP2 attenuated the foam cell formation induced by lipopolysaccharides (LPS) and oxidized low-density lipoproteins (ox-LDL) in stimulated THP-1 cells (*p* < 0.001). The expression of ATP-binding cassette transporters A (ABCA)-1, ABCG-1, scavenger receptor (SR)-B1, liver X receptor (LXR)-α, and peroxisome proliferator-activated receptor (PPAR)-γ mRNA and proteins were increased (*p* < 0.001) following MIP2 administration. A concentration-dependent decrease in the phosphorylation of p65, p38, and JNK was also observed following MIP2 administration. Moreover, an inhibition of p65 phosphorylation enhanced the expression of ABCA1, ABCG1, SR-B1, and LXR-α. TLR inhibition promoted the cholesterol efflux pathway by increasing the expression of ABCA-1, ABCG-1, and SR-B1, thereby reducing foam cell formation. Our results suggest a potential role of the p65/NF-kB/LXR-α/ABCA1 axis in TLR-mediated cholesterol homeostasis.

## 1. Introduction

Atherosclerosis, a chronic inflammatory condition, is derived from dysregulated cholesterol homeostasis with inflammatory signaling and endothelial dysfunction, leading to arterial plaque formation, a precursor to myocardial infarction and cerebrovascular accidents [1,2]. The pathogenesis involves lipoproteins undergoing modification and entrapment by vascular proteoglycans, culminating in cholesterol deposition, endothelial impairment, and an inflammatory cascade. This milieu facilitates the transformation of macrophage and smooth muscle cells into lipid-laden foam cells, marking the advent of atherosclerosis’ fatty streak phase [3].

The role of elevated serum low-density lipoproteins (LDL) as a pivotal risk factor for atherosclerotic cardiovascular disease is well established [4,5]. LDL’s transformation into an atherogenic agent occurs through oxidative modification, which, in turn, precipitates inflammatory mechanisms. Oxidized LDL (ox-LDL) not only exacerbates foam cell formation but also elicits an inflammatory milieu. Conversely, a reverse cholesterol transport (RCT) mechanism governed by an array of transmembrane proteins, including ATP-binding cassette transporters A-1 (ABCA-1) and G-1 (ABCG-1), lectin-like oxLDL (LOX-1), scavenger receptor (SR)-A, SR-B1, and cluster of differentiation 36 (CD36), orchestrate cholesterol efflux [6]. Typically, LOX-1, CD-36, and SR-A mediate the uptake of modified LDL, while ABCA-1, ABCG-1, and SR-B1 facilitate the efflux of intracellular cholesterol. Cholesterol accumulation within cells triggers lipid droplet storage and activates the liver X receptor (LXR), promoting the transcription of efflux mediators ABCA-1 and ABCG-1. Furthermore, ox-LDL has been implicated in mitochondrial oxidative stress and subsequent reactive oxygen species (ROS) production, as observed in atherosclerosis [7]. These ROS engage in lipid peroxidation reactions with membrane fatty acids, precipitating cell death via ferroptosis and generating deleterious carbonyl compounds.

Toll-like receptors (TLR) are pattern-recognition receptors that activate the innate immune system, are ubiquitously expressed by various immune cells, and engage with pathogen-related molecular patterns (PAMPs) as well as damage-associated molecular patterns (DAMPs) emanating from compromised or necrotic tissues [8,9]. TLR activation triggers a cascade of signals that culminate in the production of pro-inflammatory cytokines, chemokines, and adhesion molecules—critical elements of the innate immune defense against pathogens. Yet, persistent or excessive TLR stimulation can cause tissue damage in chronic inflammatory disorders. Among the TLRs, TLR4 is particularly instrumental in ox-LDL internalization during cholesterol homeostasis dysregulation [10].

Several molecules acting on the TLR signal pathways have been developed, some of which specifically block a particular TLR, while others nonspecifically interrupt signaling by several TLRs [11]. Recent research by Shah et al. spotlighted MIP2, a newly synthesized peptide that impedes TLR signaling pathways and has demonstrated efficacy in macrophages and mouse models of several inflammatory conditions [12]. While the molecules modulating TLR signals show differences in their functions depending on their binding site, MIP2 has not only exerted a broad-range TLR-inhibitory action but also blocked the MyD88- and the TRIF-dependent pathways of TLR4 in macrophages.

Although it has been documented that MIP2 modulates inflammation via TLR signaling inhibition, its influence on cholesterol metabolism remains unexplored. Our study aims to elucidate the interplay between TLR signaling and RCT during foam cell formation and cholesterol efflux in stimulated monocytes treated with MIP2. Additionally, we examine the regulatory potential of novel TLR inhibitors in these processes.

## 2. Results

### 2.1. Foam Cell Formation and Cholesterol Assay after MIP2 Administration

We assessed the viability of stimulated THP-1 cells treated with increasing concentrations of MIP2, as depicted in Appendix A. Notably, while the induction with LPS and ox-LDL resulted in augmented foam cell formation, the introduction of MIP2 markedly reduced the amount of foam cells (*p* < 0.001) (Figure 1). Correspondingly, both total and free cholesterol concentrations, which were elevated subsequent to LPS and ox-LDL exposure, demonstrated a significant reduction upon treatment with MIP2 (*p* < 0.001).

### 2.2. Expression of Cholesterol Efflux Transporters after MIP2 Administration

Western blot analysis showed that stimulated THP-1 cells treated with LPS alone or in combination with ox-LDL had no significant effect on the expression levels of ABCA-1, ABCG-1, SR-B1, LXRα, and PPAR-γ (Figure 2A,B). In contrast, treatment of MIP2 markedly elevated the expression of these proteins in the stimulated THP-1 cells (*p* < 0.001). Interestingly, the expression levels of ABCA-1, ABCG-1, and PPAR-γ significantly increased with the administration of 10, 20, and 30 μM MIP2 (*p* < 0.001), while the expression levels of SR-B1 and LXRα were notably enhanced with the administration of 20 and 30 μM MIP2 (*p* < 0.01).

Furthermore, RT-PCR analysis corroborated a significant upregulation in the expression of ABCA-1, ABCG-1 SR-B1, LXRα, and PPAR-γ following MIP2 treatment in the stimulated THP-1 cells (Figure 2C). The expression levels of ABCA-1 and ABCG-1 rose with 20 and 30 μM MIP2 administration (*p* < 0.001), and the expression levels of SR-B1, LXRα, and PPAR-γ significantly increased upon treatment with 10, 20, and 30 μM MIP2 (*p* < 0.001).

### 2.3. Change in Inflammatory Cytokines after MIP2 Administration

Exposure to LPS alone or in combination with ox-LDL resulted in elevated levels of IL-1β, TNFα, IL-6, and IL-10 in stimulated THP-1 cells (Figure 3). However, the administration of MIP2 at 10, 20, and 30 μM led to a dose-dependent reduction in the levels of TNFα and IL-1β (*p* < 0.01). Conversely, the levels of IL-6 and IL-10 were minimally affected or remained undetectable following treatment with MIP2 at any of tested concentrations in the stimulated THP-1 cells (*p* < 0.01).

### 2.4. Change in Phosphorylation of Intracellular Transcription after MIP2 Administration

LPS treatment or the combination of LPS and ox-LDL induced phosphorylation of p65, which was reduced in a dose-dependent manner upon administration of MIP2 at the concentrations of 10, 20, and 30 μM (*p* < 0.001) (Figure 4A,B). The phosphorylation of IκBα triggered by LPS and LPS + ox-LDL treatments was inhibited by MIP2 at the same concentrations (*p* < 0.001). Similarly, phosphorylation of JNK and p38, also induced by LPS and LPS + ox-LDL, was suppressed by MIP2 administration at 10, 20, and 30 μM (*p* < 0.001).

Additionally, the study explored the impact of transcription factor inhibitors on the expression of cholesterol efflux receptors. The administration of Bay 11-7082, an NF-κB inhibitor, increased the expression of ABCA1, LXRα, and SR-B1 in stimulated THP-1 cells with the treatment of LPS or LPS + ox-LDL (*p* < 0.001) (Figure 4C,D). The expression levels of ABCG1 and PPAR-γ were also elevated by administration of Bay 11-7082 in stimulated THP-1 cells with the treatment of LPS + ox-LDL (*p* < 0.001). However, the use of SP (a JNK inhibitor) and SB (a p38 inhibitor) did not alter the expression of these genes.

### 2.5. ROS Production after MIP2 Administration

In the stimulated THP-1 cells, ROS production was elevated following the treatment of LPS or LPS combined with ox-LDL. The introduction of 10 μM MIP2 did not modify ROS levels. However, a significant reduction in ROS production was observed in stimulated THP-1 cells treated with LPS and ox-LDL when 20 and 30 μM MIP2 were administered (*p* < 0.01) (Figure 5).

## 3. Discussion

Our findings indicate that treatment with a TLR antagonist in monocytes exposed to LPS and ox-LDL, which are known to induce inflammatory stimulation and uptake of modified LDLs, respectively, resulted in a reduction in foam cell formation. The expression levels of ABCA-1, ABCG-1, and SR-B1 were elevated in TLR-antagonist-treated stimulated monocytes compared to those not receiving TLR antagonist treatment. Additionally, a decrease in the phosphorylation of p65, p38, and JNK was observed in the TLR antagonist-treated cells. Notably, inhibiting the phosphorylation of p65 further enhanced the expression of ABCA-1, ABCG-1, SR-B1, and LXRα. The administration of LPS and ox-LDL serves as mimic of the inflammatory and proatherogenic conditions associated with prolonged chronic inflammatory disorders in humans, including rheumatic diseases. Our results suggest that TLR antagonist treatment mitigates foam cell formation by upregulating the intracellular transcription signals that govern cholesterol efflux in stimulated monocytes. It is known through the constitutional data that inflammation can be regulated by modulating TLR signal, but this study revealed that modulating TLR signal results in improving cholesterol efflux and reducing foam cell formation and oxidative stress, which are important in atherosclerosis [11,13]. It implies that when the therapies blocking TLR signaling are used in inflammatory diseases, they can be promised to prevent the atherosclerosis, which can be accelerated due to an inflammatory response or the use of drugs.

MIP2 inhibits a broad range of TLRs, including TLR-2 and -4, thereby lowering the secretion of inflammatory cytokines and blocking the TLR-dependent signaling pathway [12]. Previous studies have reported the role of MIP2 in mitigating glomerular proliferation with cell infiltration in a lupus-prone mouse model. Similarly, MIP2 has been shown to reduce plasma levels of IL-1β, IL-6, and TNFα in an LPS-induced sepsis mouse model and to alleviate liver steatosis and macrovesicular fat changes in a mouse model of nonalcoholic steatohepatitis. Our findings suggest that LPS and ox-LDL increase the levels of TNFα, IL-1β, IL-6, and IL-10 in stimulated monocytes, while MIP2 treatment decreases levels of TNFα and IL-1β in a dose-dependent manner and blocks the increase in levels of IL-6 and IL-10 at even minimal doses.

The TLR signal pathway is a promising target for treating inflammatory disorders, including sepsis and sterile inflammatory diseases [14]. Antagonists targeting TLR signaling can block excessive immune activation by sustained pro-inflammatory cytokines and chemokines [15,16]. Several TLR signals are involved in both the inflammatory response and cholesterol metabolism. Studies have highlighted the roles of TLR4 and TLR2 in foam cell formation and cholesterol efflux [17,18,19]. Exogenous TLR2 treatment has been found to decrease cholesterol efflux by reducing ABCA-1, ABCG-1, and SR-B1 expression in a dose-dependent manner [20]. TLR2 contributes to atherosclerosis progression in LDL receptor knock-out (KO) mice, and inactivation of the TLR2 gene has been associated with reduced atherosclerosis progression in ApoE KO mice [21,22]. Deficiencies in TLR4 or MyD88 have been shown to reduce the sizes of atherosclerotic lesions in mouse models, with the TLR4-dependent signaling pathway promoting ox-LDL uptake and subsequent intracellular lipid accumulation in macrophages [10,23]. TLR4 deficiency reduces lipid accumulation in ApoE-KO mice, and TLR4 antibodies block the differentiation of TLR4-competent macrophages into foam cells [19,24]. Therefore, inhibition of the TLR signaling pathway may control pro-inflammatory and pro-atherosclerotic conditions in chronic inflammatory disorders.

PPARs and LXRs are nuclear receptors that interact with TLRs and PAMPs/DAMPs to regulate the cholesterol efflux pathway. In an atherosclerosis-prone mouse model, activation of PPAR-γ induced LXRα expression, whereas defective PPAR-γ promoted foam cell formation from macrophages. Similarly, the activation of TLR by microbial ligands has been shown to inhibit LXRα induction [25,26,27]. Decreased expression of LXRα has been associated with the downregulation of ApoE, ABCA-1, and ABCG-1. Propofol administration has increased PPAR-γ and LXRα expression along with upregulation of ABCA-1, ABCG-1, and SR-B1 in THP-1 cell-derived foam cells [28,29]. While cholesterol loading activates LXRα, defects in the intracellular LXRα pathway lead macrophages into pro-inflammatory states with increased cholesterol accumulation, resulting in foam cell formation [30]. Our data confirm the role of TLR signaling in regulating the transcription of several cellular receptors/cholesterol transporters essential for cholesterol efflux. Moreover, blocking the TLR signal downregulates p65 phosphorylation, which activates the PPAR-γ- LXRα- ABCA-1/ABCG-1/SR-B1 pathway.

We also observed that the TLR antagonist downregulated LPS and ox-LDL-induced ROS production, particularly at higher doses. Ox-LDL stimulates neutrophil-mediated ROC production by activating NADPH oxidase [31,32]. LPS-induced TLR4 activation increases ROS production through the phosphorylation of AKT, PKC and p38―signals that are inhibited by specific TLR4 inhibitors [33]. TLR4 and TLR2 induce ROS generation by interacting with IL-1 receptor-associated kinase (IRAK), extracellular signal-regulated kinase, and Nox2 in monocytes [34]. We have demonstrated that MIP2 decreased IL-1β concentrations in a dose-dependent manner. Blocking TLR through MIP2 could inhibit the IRAK–Nox2 signaling pathway, downregulating ROS production in the monocytes stimulated with LPS and ox-LDL.

Our study has limitations. We did not investigate the effect of TLR inhibition on atherosclerosis progression. Although vascular changes were not assessed, we elucidated alterations in receptor expressions involved in cellular cholesterol metabolism. Further pathohistological changes in arteries could be explored through the knockout of TLR or intracellular receptors in an atherosclerosis-prone animal model. Since TLR activation occurs under various conditions, including infection and inflammatory diseases, efforts are being made to modulate TLR signaling for therapeutic purposes. Our study confirmed that blocking TLR enhanced the expression of cholesterol efflux receptors and downregulated foam cell formation though the TLR/NF-κB/p-65 axis in stimulated monocytes.

## 4. Materials and Methods

### 4.1. Cell Culture and Reagents

Human THP-1 cells were cultured in Roswell Park Memorial Institute (RPMI)-1640 medium (Gibco, Grand Island, NY, USA) supplemented with penicillin (100 U/mL) and 10% fetal bovine serum (Thermo Fisher Scientific, Inc., Rockford, USA). Cells were incubated in a humidified incubator at 37 °C with 5% CO_2_. THP-1 cells (1 × 10^6^ cells/mL) were differentiated into macrophages using 20 nM phorbol myristate acetate (PMA; Sigma-Aldrich, St. Louis, MO, USA) in RPMI-1640 medium for 24 h. THP-1 macrophages were pre-treated with 10–30 μM MIP2 (Peptron, Daejon, Korea) and exposed to human ox-LDL (40 μg/mL) or LPS (100 ng/mL). MIP2 was synthesized using Peptron at a purity of 97.17% as previously described [12].

### 4.2. Oil Red-O Staining

THP-1 cells were washed with phosphate buffered saline (PBS) containing 0.05% Tween-20 and fixed with 4% paraformaldehyde for 5 min. The cells were stained with 0.2% Oil Red-O (Sigma-Aldrich) in 60% isopropanol for 30 min at room temperature. The images were obtained using a light microscope and the foam cell formation was analyzed. Oil Red-O was quantified by dissolving the stained cells in 100% isopropanol and measuring the absorbance at a wavelength of 500 nm.

### 4.3. Cholesterol Assay

Differentiated THP-1 cells (1 × 10^6^ cells/mL) were stimulated with LPS and ox-LDL in the presence of MIP2 for 24 h. Cells were harvested in chloroform/isopropanol/NP-40 (7:11:0.1) and centrifuged for 10 min at 15,000× *g*. The liquid phase was transferred into a centrifuge tube and air-dried at 50 °C to remove any remaining solvent. The dried lipids were dissolved in a cholesterol assay buffer by sonication. Total and free cholesterol were quantified using the EZ-Total Cholesterol Assay Kit (DoGenBio, Seoul, Korea) according to the manufacturer’s instruction.

### 4.4. Western Blotting

The cells were lysed with RIPA lysis buffer containing 1 mM phenylmethylsulfonyl fluoride (PMSF). Equal amounts of extracted proteins were separated on 10% SDS–PAGE gels and transferred onto polyvinylidene fluoride (PVDF) membranes for incubation with specific primary antibodies (1:1000–1:2000) at 4 °C overnight. Antibodies against p-JNK, JNK, p-p38, p38, p-IκBa, IκBa, p-p65, p65, ABCA-1, ABCG-1 and PPAR-γ were purchased from Cell Signaling Technology Inc. (Danvers, MA, USA); against ABCG-1 from Proteintech (Rosemont, IL, USA); against SR-B1 and LXR-α from Novus (Centennial, CO, USA). A peroxidase-conjugated anti-mouse or anti-rabbit antibody was used as the secondary antibody. Bands were visualized using enhanced chemiluminescence reagents and recorded using the LAS-4000 luminescent image analyzer (Fujifilm, Stamford, CT, USA).

### 4.5. Quantitative Real-Time PCR

Total RNA was extracted from the macrophages using Minibest Universal RNA Extraction Kit (Takara Bio Inc.; Otsu, Shiga, Japan). cDNA was prepared from 2 μg RNA with the PrimeScript RT reagent kit (Takara Bio Inc.; Otsu, Shiga, Japan). Real-time PCR was performed using the SYRB Premix Ex Taq kit (Takara Bio Inc.; Otsu, Shiga, Japan). The primer sequences were for ABCA-1, forward, 5′-GAAGTACATCAGAACATGGGC-3′ and reverse, 5′-GATCAAAGCCATGGCTGTAG-3′; ABCG-1, forward, 5′-CCCTCAGAATGCCAGCAGTT-3′ and reverse, 5′-CCGAGACACACACCGACTTG-3′; LXR-α forward, 5′-TCTGGAGACATCTCGGAGGTACAAC-3′ and reverse, 5′-AGCAAGGCAAACTCGGCATC-3′; PPAR-γ forward, 5′-CCTCCCTGATGAATAAAGATGG-3’ and antisense, 5′-GCAAACTCAAACTTAGGCTCCA-3′; SR-B1 forward, 5′-CCTTGTTCCTGGACATCCAC-3′, reverse, 5′-CTCAATCTTCCCAGTTTGTCCA-3′; GAPDH forward, 5’-TGCCATCAACGACCCCTTCA-3′ and reverse, 5’-TGACCTTGCCCACAGCCTTG-3′. The 2^−ΔΔCt^ method was used to assess the relative mRNA expression level normalized to that of GAPDH.

### 4.6. Enzyme-Linked Immunosobent Assay (ELISA)

To analyze the effect of MIP2 on cytokine production in ox-LDL-stimulated THP-1 macrophages, cell-free supernatants were collected and the levels of IL-1β, IL-6, IL-10 and TNF-α were determined using ELISA kits (R&D systems, Minneapolis, MN, USA) following the manufacturer’s instruction.

### 4.7. ROS Production

The differentiated cells (1 × 10^6^ cells/mL) were grown overnight. After MIP2 treatment and LPS or oxidized LDL stimulation, ROS levels were quantified by DCDF-DA (2’,7’-dichloroie-fluorescein diacetate; Thermo Fisher Scientific, Inc., Rockford, IL, USA).

### 4.8. Statistical Analysis

All experiments were repeated at least three times. Data are presented as mean ± S.E.M and analyzed by unpaired *t*-test and one-way ANOVA with LSD post hoc test. *p* < 0.05 is considered statistically significant.

## 5. Conclusions

In this study, a TLR antagonist was found to reduce foam cell formation in monocytes treated with LPS and ox-LDL and to increase expression of LXRα, PPAR-γ, ABCA-1, ABCG-1, and SR-B1. Additionally, the TLR antagonist inhibited the phosphorylation of p65, p38, and JNK. The inhibition of p65 phosphorylation was shown to enhance the expression of LXR-α, ABCA-1, ABCG-1, and SR-B1.

## Figures and Tables

**Figure 1 ijms-25-06808-f001:**
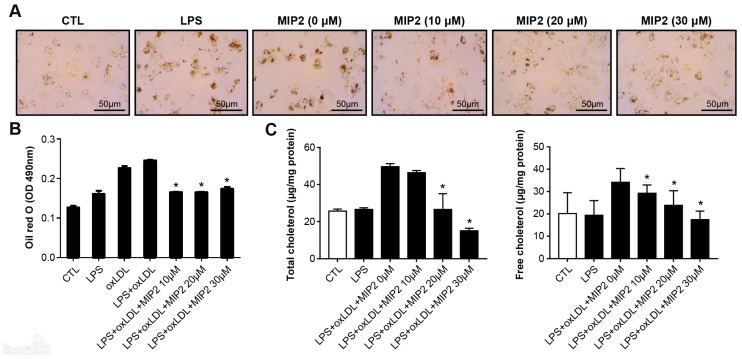
Changes in foam cell formation and cholesterol levels in stimulated THP-1 cells with LPS and oxidized LDL after MIP2 treatment. (**A**,**B**). Oil red-O stainings show Foam cells are increased in stimulated THP-1 cells with LPS and LPS + oxidized LDL, and decreased in those after administration of MIP2 10, 20, and 30 μM (* *p* < 0.001). (**C**) Amount of total cholesterol is increased in stimulated THP-1 cells with LPS + oxidized LDL and those after administration of MIP2 10 μM and decreased in those after administration of 20 and 30 μM MIP2 (* *p* < 0.001). Amount of free cholesterol is increased in stimulated THP-1 cells with LPS + oxidized LDL and decreased in those after administration of MIP2 10, 20, and 30 μM (* *p* < 0.001).

**Figure 2 ijms-25-06808-f002:**
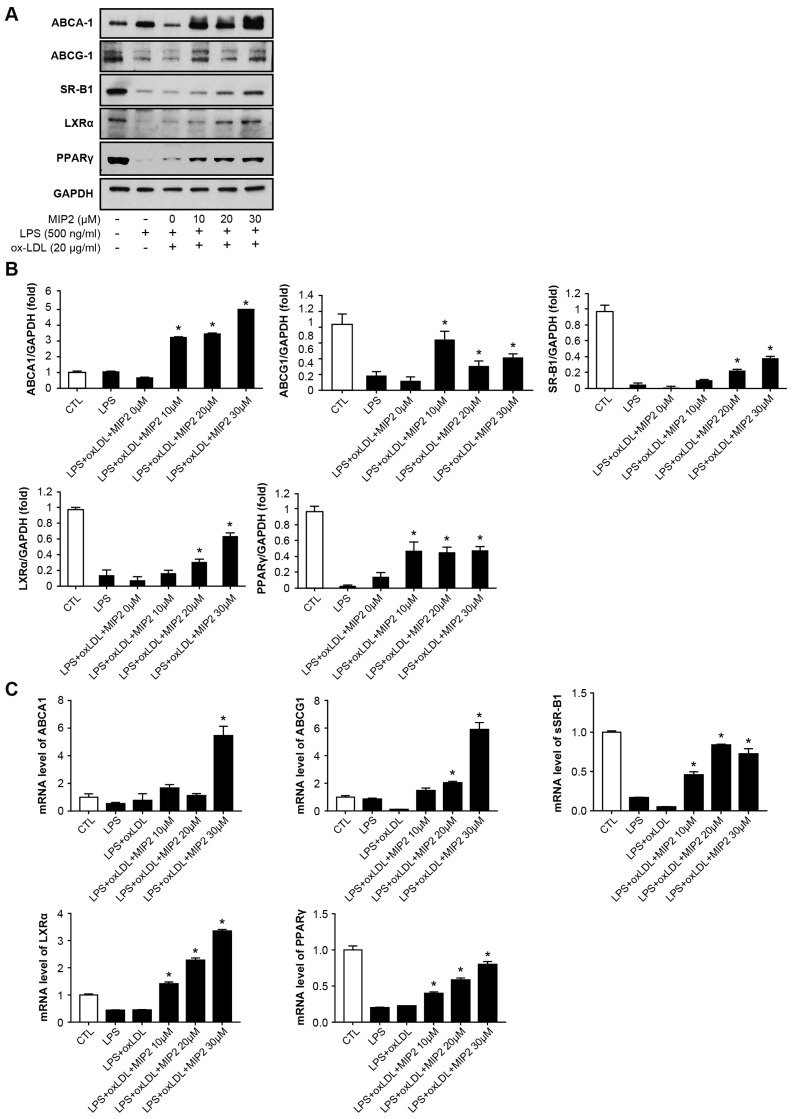
Changes in cholesterol efflux-associated receptors in stimulated THP-1 cells with LPS and oxidized LDL after MIP2 treatment. (**A**,**B**). WB results show that expressions of ABCA1 and ABCG1 do not differ in stimulated THP-1 cells with LPS + oxidized LDL after administration of 10, 20 and 30 μM MIP2 (* *p* < 0.001), expressions of SR-B1 and LXRα are increased in those after administration of 20 and 30 μM MIP2 (* *p* < 0.001), and expressions of PPARγ are increased in those after administration of MIP2 10, 20, and 30 μM (* *p* < 0.001). (**C**) RT-PCR results show that mRNA levels of ABCA1 are increased in the stimulated THP-1 cells with LPS + oxidized LDL after the administration of MIP2 30 μM (* *p* < 0.001), mRNA levels of ABCG1 are increased in the stimulated THP-1 cells with LPS + oxidized LDL after administration of 20 and 30 μM MIP2 (* *p* < 0.001), and mRNA expressions of SR-B1, LXRα, and PPARγ are increased in the stimulated THP-1 cells with LPS + oxidized LDL after administration of MIP2 10, 20, and 30 μM (* *p* < 0.001).

**Figure 3 ijms-25-06808-f003:**
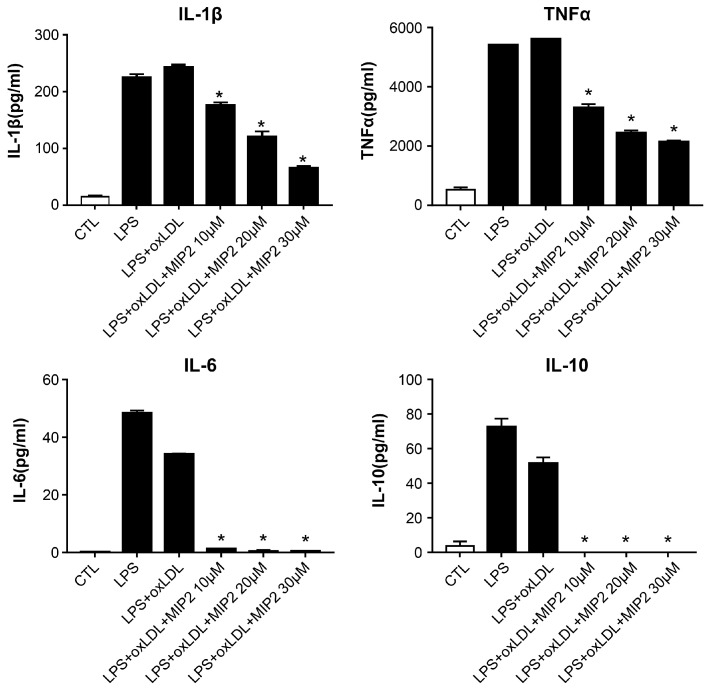
Changes in cytokines levels in stimulated THP-1 cells with LPS and oxidized LDL after MIP2 treatment. Concentrations of IL-1β, TNF-α, IL-6, and IL-10 were decreased in the stimulated THP-1 cells with LPS and oxidized LDL after the administration of 10, 20 and 30 μM MIP2 (* *p* < 0.001).

**Figure 4 ijms-25-06808-f004:**
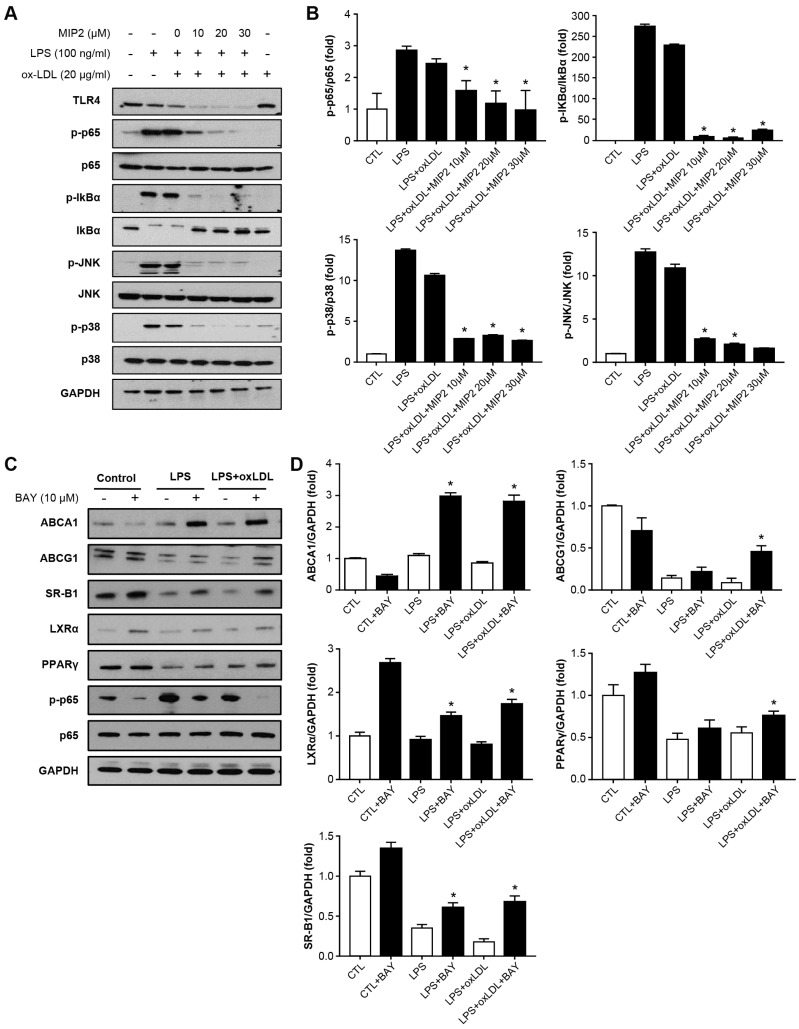
Changes in phosphorylation of MAP kinases in stimulated THP-1 cells with LPS and oxidized LDL after MIP2 treatment. (**A**,**B**). WB results show that expressions of phosphorylated p65, p38, and JNK are decreased in stimulated THP-1 cells with LPS and oxidized LDL after administration of 10, 20 and 30 μM MIP2 (* *p* < 0.001). (**C**,**D**) WB results show that expressions of ABCA1, ABCG1, SR-B1, LXRα and PPARγ are increased in stimulated THP-1 cells with LPS and oxidized LDL after BAY treatment (* *p* < 0.001).

**Figure 5 ijms-25-06808-f005:**
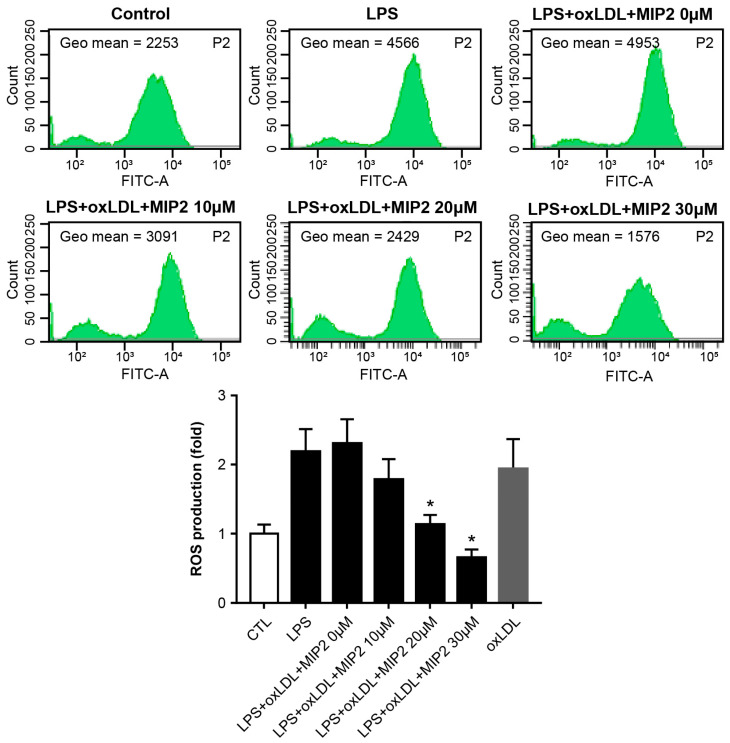
Changes in reactive oxygen species (ROS) production in stimulated THP-1 cells with LPS and oxidized LDL after MIP2 treatment. Amount of ROS production are decreased in stimulated THP-1 cells with LPS + oxidized LDL after administration of 20 and 30 μM MIP2 (* *p* < 0.001).

## Data Availability

All data cited in the study are publicly availble.

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
