# Peer review of "Inhibition of Toll-like Receptors Alters Macrophage Cholesterol Efflux and Foam Cell Formation"

_ijms, 2024, doi:10.3390/ijms25126808_

Round 1

Reviewer 1 Report

Comments and Suggestions for Authors

Dear authors

The manuscript explains with biochemical experiments the “Inhibition of toll-like receptors alters macrophage cholesterol efflux and foam cell formation”

 Here are my suggestions for improvement the manuscript:

Lines 30-32: In Atherosclerosis only cholesterol is important? You could also refer, very shortly, to other parameters that contribute to atherosclerosis. I believe that you should do this because you are in the section of the “Introduction”.

Line 61:Recent search by Shah et al. spotlighted MIP2…” I think that you start in a different paragraph and describe in one line about inhibitors with similar characteristics and then describe your inhibitor.

Lines 171-172: You “…..support the potential therapeutic strategy of targeting TLR signaling pathways………” You could explain in more detail if exist other therapeutic strategies exist so far for that purpose? What are the differences with yours?

Lines 233-235: “These findings suggest that the inhibition of TLR signaling holds promise for preventing atherosclerosis, and therapeutics targeting this signaling might be beneficial for preventing and treating cardiovascular diseases”. I believe there is a distance between inhibition of TLR signaling and therapeutic targeting. I believe you should write that in a different way.

Line 310: “In this study, a newly synthesized TLR antagonist called MIP2 was found to reduce” Here you conclude the meaning of a newly synthesized TLR antagonist. Previously you don’t describe it at al.

Author Response

We appreciate your review of our manuscript “Inhibition of toll-like receptors alters macrophage cholesterol efflux and foam cell formation”. In response to your comments and those of the reviewers, we have made several changes to the text, as summarized below.

  1. Lines 30-32: In Atherosclerosis only cholesterol is important? You could also refer, very shortly, to other parameters that contribute to atherosclerosis. I believe that you should do this because you are in the section of the “Introduction”.

Answer) Thank you for the comment. We agreed with your points of view, modified the sentence briefly according to the “Introduction”, and underlined it in the revised manuscript.

  1. Line 61: “Recent search by Shah et al. spotlighted MIP2…” I think that you start in a different paragraph and describe in one line about inhibitors with similar characteristics and then describe your inhibitor.

Answer) Thank you for the comment. We described the molecules actively being developed that act on TLR signals up to recent times and the strengths of MIP2 with the associated reference ([10]. We underlined them (Line 63 – 70) in the revised manuscript.

  1. Lines 171-172: You “…..support the potential therapeutic strategy of targeting TLR signaling pathways………” You could explain in more detail if exist other therapeutic strategies exist so far for that purpose? What are the differences with yours?

Answer) Thank you for the comment. Our data showed that TLR signal modulation can not only reduce inflammation but also enhance intracellular cholesterol efflux. Those effects into cholesterol metabolism could prevent atherosclerosis, which is accelerated in chronic inflammatory diseases, when drugs blocking TLR signals are used in inflammatory disease in the future. We added the sentences and references, and underlined them in the revised manuscript.

  1. Lines 233-235: “These findings suggest that the inhibition of TLR signaling holds promise for preventing atherosclerosis, and therapeutics targeting this signaling might be beneficial for preventing and treating cardiovascular diseases”. I believe there is a distance between inhibition of TLR signaling and therapeutic targeting. I believe you should write that in a different way.

Answer) Thank you for the comment. We understood that you found those statements to be somewhat exaggerated and difficult to accept. Therefore, we decided to exclude the content that expands our data.

  1. Line 310: “In this study, a newly synthesized TLR antagonist called MIP2 was found to reduce” Here you conclude the meaning of a newly synthesized TLR antagonist. Previously you don’t describe it at al.

Answer) Thank you for the comment. We described MIP2 as a newly synthesized TLR antagonist in Introduction (Line 65-67), modified the last phrase of the discussion (Line 312-314), and underlined them in the revised manuscript.

Reviewer 2 Report

Comments and Suggestions for Authors

I would like to begin by congratulating the authors on their study and manuscript. I find their work very interesting and well-presented as well as a great applicability for the subject in future medical practice.

Therefore i consider that the manuscript is suitable for publication in its current form.

Author Response

Thank you for your comment. We hope that the manuscript is published in this jorunal.

Reviewer 3 Report

Comments and Suggestions for Authors

Peer review: “Inhibition of toll-like receptors alters macrophage cholesterol efflux and foam cell formation”.

Jae-Mi Kim et Al’s study precisely illustrates the interplay between toll-like receptors signaling and reverse cholesterol transport during foam cell formation. It also deepens cholesterol efflux process in monocytes stimulated with tool-like receptors antagonists and examine their regulatory potential in these processes.

The manuscript is well articulated and results are clearly reported. The authors use a specialized and technical language and they cite all their sources, including a complete list of bibliographic references.

Here below some minor comments:

ABSTRACT : The abstract is well written, it matches the content of the paper and effectively summarize the manuscript.

INTRODUCTION: The introduction provides useful background information on the research topic. Here some minor comments for the authors:

- Rows 38-48: bibliographic references missing. Please add references to support the statements included from row 38 till row 48.

RESULTS: This section precisely describes the results of the study. The authors use a technical language and show a deep knowledge of the research area. However, figures, although quite accurate, might be a little clearer to make results easier to understand. Please consider making some changes.

Here below some comments/suggestions for the authors.

- Figures 1 and 2. Please consider putting the different panels on different lines - for example – in order to make them easier to read:

A-Panel

B-Panel

C-Panel

- Row 141: Fig 3 should be Fig 4.

Moreover in this figure D label (for D panel) is missing.

MATERIALS AND METHODS: No comments to add

DISCUSSION/CONCLUSIONS: The discussion section fits very well with the aims of the study as stated in the introduction. Minor suggestion:

- Please consider to be more concise and summarize more the results in order to provide an easier overview to the readers.

BIBLIOGRAPHY: Bibliography section is complete and accurate.

Comments on the Quality of English Language

English style is fine

Author Response

We appreciate your review of our manuscript “Inhibition of toll-like receptors alters macrophage cholesterol efflux and foam cell formation”. In response to your comments and those of the reviewers, we have made several changes to the text, as summarized below.

  1. ABSTRACT: The abstract is well written, it matches the content of the paper and effectively summarize the manuscript.
  2. INTRODUCTION: The introduction provides useful background information on the research topic. Here some minor comments for the authors:

- Rows 38-48: bibliographic references missing. Please add references to support the statements included from row 38 till row 48.

Answer) Thank you for the comment. Although, we describe rows 38-48, which are derived from references 4 and 5, we added one more reference as Ref.6 and underlined in the revised manuscript.

  1. RESULTS: This section precisely describes the results of the study. The authors use a technical language and show a deep knowledge of the research area. However, figures, although quite accurate, might be a little clearer to make results easier to understand. Please consider making some changes.

Here below some comments/suggestions for the authors.

- Figures 1 and 2. Please consider putting the different panels on different lines - for example – in order to make them easier to read:

A-Panel

B-Panel

C-Panel

- Row 141: Fig 3 should be Fig 4.

Moreover in this figure D label (for D panel) is missing.

Answer) Thank you for the comment. We corrected them as you commented.

  1. MATERIALS AND METHODS: No comments to add
  2. DISCUSSION/CONCLUSIONS: The discussion section fits very well with the aims of the study as stated in the introduction. Minor suggestion:

- Please consider to be more concise and summarize more the results in order to provide an easier overview to the readers.

Answer) Thank you for the comment. We corrected some sentences as you commented and underlined them in the revised manuscript.

  1. BIBLIOGRAPHY: Bibliography section is complete and accurate.

Thank you for the constructive review. We hope that the revised manuscript now meets the journal’s standards for publication.